# Trajectory Tracking Control Study of Unmanned Fully Line-Controlled Distributed Drive Electric Vehicles

**Tian Tian [1], Gang Li [1,*], Yuzhi Li [1], Ning Li [2] and Hongfei Bai [1]**

[1] School of Automobile and Traffic Engineering, Liaoning University of Technology, Jinzhou 121001, China; 219904017@stu.lnut.edu.cn (T.T.); baihf@lnut.edu.cn (H.B.)

[2] School of Electronics & Information Engineering, Liaoning University of Technology, Jinzhou 121001, China; dzxxln@lnut.edu.cn

[*] Correspondence: qcxyligang@lnut.edu.cn

**Abstract:** Unmanned fully line-controlled distributed drive electric vehicles with four-wheel independent drive, dependent braking and dependent steering have significant advantages over conventional vehicles in terms of dynamic control, but at the same time multiple actuators with multiple degrees of freedom also pose the risk of failure in the steering system, which is studied in this paper for trajectory tracking control. Rational control of multiple systems such as drive, braking, steering and fault tolerance of the unmanned fully line-controlled distributed drive electric vehicles are carried out. For longitudinal control, a fuzzy PI algorithm is used to input velocity error and velocity error rate of change, and to solve the required drive torque of the vehicle based on fuzzy rules; for lateral control, according to model prediction control theory, the exact model is predicted and an optimized search is performed to reasonably allocate the forward and backward wheels turning corners ensuring the accuracy and roadholding of trajectory tracking; for fault-tolerant control, differential drive and other methods of control, when a fault is detected, the number and position information of the faulty steering motor is transmitted to the fault-tolerant decision module, which outputs control commands according to the decision. The outcomes demonstrate that the presented trajectory following the control policy enhances the precision, roadholding and safety of trajectory following in an effective way.

**Keywords:** unmanned driving; fully line-controlled; distributed-drive electric vehicles; trajectory tracking control; fault-tolerant control





## 1. Introduction

The four-wheel drive, braking force and steering angle can be independently controlled with multi-degrees of freedom, and the full linear control distributed-drive electric automobile has become the new energy intelligent electric vehicle research hot spot, but it may bring the risk of actuator failure. When some of the actuators fail, the reasonable control of the fully line-controlled distributed-drive electric vehicle to guarantee the precision, stability and safety of traceability has become a key topic of research. The technology of trajectory tracking control is critical for the realization of unmanned fully line-controlled distributed drive electric vehicles. With the speedy development of unmanned fully line-controlled distributed drive electric vehicles' technology, the questions of accuracy, stability as well as the safety of trajectory tracking control have gained wide attention.

In this area, numerous investigations have been carried out in various countries, including pure pursuit [1,2], Stanley [3], sliding mode [4,5], adaptive [6,7], model predictive [8,9], nonlinear model predictive control [10], potential field control [11–13], and feed-forward pre-scanning control [14]. Matthew Brown et al. conceived of a comprehensive control architecture for path planning and tracking control by MPC theory to enhance the security and trajectory following precision of the autonomous car [15]. Bobier applied a sliding mode variable construction control technique to provide fast and stable trajectory

tracking by estimating the lateral oscillation angular velocity. [16]. Matteo Corno et al. introduced an LPV technique for automatic route-following in the face of steering drive nonlinearity, which successfully reduced tracking errors [17]. Y. H. Li et al. designed adaptive time-domain parameter algorithms for trajectory tracking controllers, and simulation experiments showed that they improved the maneuvering stability, trajectory tracking accuracy and safety performance of self-driving vehicles [18]. An EMPC with a flexible artificial potential field was created by Hongjiu Yang et al. which implemented the automatic electric vehicle to avoid obstacles and trajectory tracking, and experimental results indicated that the designed controller enhanced the effectiveness of obstacle avoidance and track following and ensured the system reliability [19]. Xinxin Liu et al. investigated the trajectory of an independent four-wheel drive steering system's tracking control machine at high speeds and proposed a feedforward, feedback and a dynamic constraint-based MPC algorithm to improve high-speed trajectory tracking performance [20]. Xin Fan et al. conducted a study on track following control for a driverless-type wheeled tractor using an upgraded quantum genetic LQR method. The results of the simulation demonstrated that the tractor has excellent tracking precision as well as stability [21]. Can Yang et al. employed MPC and fuzzy PID-based controllers for intelligent driving vehicle trajectory-tracking control, and the simulated results suggest that the controller has outstanding behavior [22]. Linhe Ge et al. coupled vertical and horizontal control of the driverless car using an offset-free MPC algorithm to realize improved trajectory-following precision and high-speed reliability [23]. Zhiwei He et al. proposed a two-level controller for horizontal trail-following control for self-driving cars, using an upper level continuous that changed over time, an MPC controller and a function with a radial basis proportional-integral derivation neural network controller in the lower level. The calculation showed that the designed controller can ensure car reliability and good trajectory-following accuracy [24]. Teng Li et al. presented a track tracking control algorithm for the smart electric automobile according to the adaptive optimal control of weight coefficients, which effectively reduced tracking errors through experimental verification [25]. A double neural internetwork active fault control strategy based on the feedforward offset technique of neural network estimation was proposed by Jian Hu et al. Both emulation and experimentation leads prove that the control strategy has good capability [26]. Jian Ou et al. proposed a longitudinal and lateral force co-reconfiguration fault-tolerant controllers policy according to active turn control, which was shown to significantly improve vehicle reliability of driving and security in the event of an actuator failure [27]. Based on deep reinforcement learning theory, Huifan Deng suggested a fault-tolerant control system for four-wheel drive electric cars. Vehicle stability, driver comfort and power consumption were effectively improved [28]. Ding, Z. et al. designed an improved dynamic Kalman filter and proposed a model predictive control method considering the anti-tilt constraint. The analogue findings indicate the approach's utility [29]. Wang T proposed a hierarchical plan and follow-up frame for self-driving cars. The simulation results show the ability to solve most common scenarios on structured roads with reasonable decision and control capabilities [30].

One of the trends in the development of intelligent vehicles today is to equip them with more sensors, particularly in the case of electric vehicles. These sensors provide a wealth of data and information that can be used for vehicle control and driver assistance systems. However, ensuring synchronization of different sensors in terms of time and space is crucial for accurate vehicle control signal input. There are ongoing research efforts to address this issue. For example, [31] is a study that uses deep learning techniques to detect tassels in crops using data from multiple sensors and transfer learning to improve detection accuracy.

To conclude, there currently exist a number of established methods for track-following management of typical low-speed circumstances, but there are numerous crises in complex traffic scenarios, such as wire-controlled steering system failure, etc. The proposed trajectory tracking-control strategy addresses the problem of ensuring track-following precision and reliability in the track-following control process of unmanned cars, while fully considering

the impact of a malfunctioning wire-controlled turning system on vehicle trajectory-tracking driving, designing wire-controlled steering fault-tolerant techniques and the validity and robustness of the strategies are also validated through experimentation.

In this article, we examine the reasonable control of the braking, driving, steering and fault-tolerant systems when some actuators fail to achieve the lability of track-following precision security. The complete vehicle control strategy for trajectory tracking control with fault-tolerant control of an unmanned fully line-controlled distributed drive electric vehicle is displayed in Figure 1.

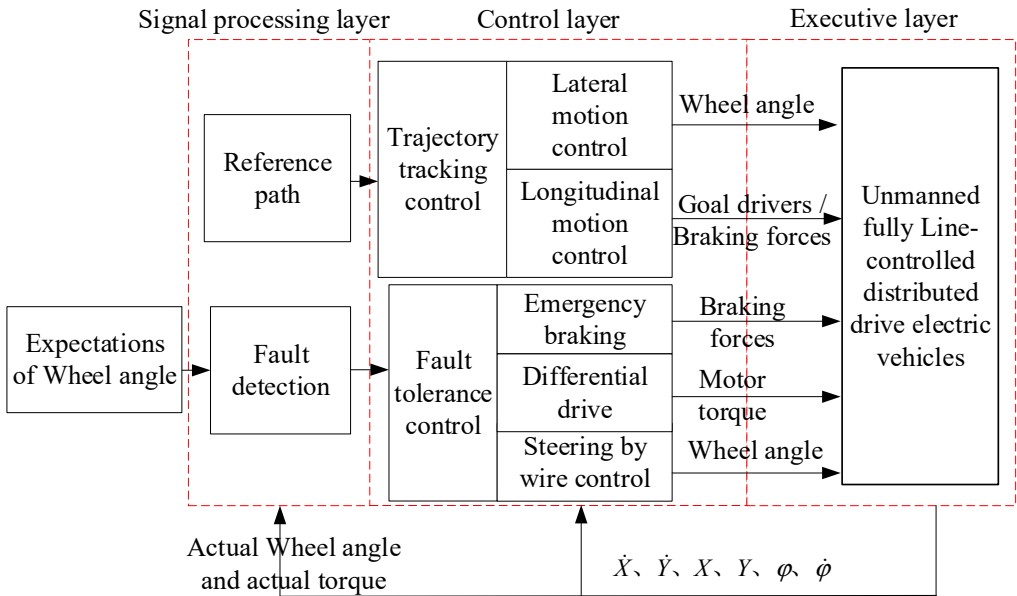

**Figure 1.** Complete vehicle control strategy.

As shown in Figure 1, $\dot{X}$ for vertical velocity, $\dot{Y}$ for horizontal velocity, $X$ for vertical displacement, $Y$ for horizontal displacement, $\varphi$ is yaw, $\dot{\varphi}$ is yaw rate.

The whole vehicle is controlled in layers, with a signal processing layer, control layer and executive layer. Signal processing layer: according to the collected information on the exact status of the vehicle at a certain time by sensors and the planned reference route for the variances between the two sources of information, send the results to the control layer. The fault diagnosis module also detects the status of the steering-by-wire sensors and actuators and sends the results to the control layer.

The thesis focuses on the trajectory tracking control and fault-tolerant control of an unmanned fully line-controlled distributed drive electric vehicle, so the track-following control at the control level consists mainly of transverse motion control and vertical motion control. The fault-tolerant control transmits the malfunctioning message gained by the malfunctioning detection module to the fault-tolerant decision module for discrimination and transmits the decision command to the unmanned fully line-controlled distributed drive electric vehicle actuator for fault-tolerant control.

## 2. Horizontal Control Strategy

The drive control strategy of the unmanned fully line-controlled distributed drive electric vehicle is to indirectly control the vehicle speed by means of controlling the drive motors' drive torque and thus the current speed tracks the expected speed. When the speed of the vehicle is faster, the computational efficiency of the controller has greater requirements; thus, for the purpose of minimizing the calculation workload and optimizing the live control policy, the fuzzy PI algorithms are applied. The velocity bias and the change rate of the speed bias are used as inputs to the fuzzy PI controller, which solves the drive torque required for an unmanned fully wire-controlled distributed-drive electric

vehicle according to fuzzy rules. The drive torque is applied to the four drive motors and distributed to the four wheels for acceleration control.

Braking control is required when the actual speed exceeds the desired speed. A braking model is established and the deceleration is calculated in accordance with the deviation from the actual velocity and the desired one and converted to the desired braking force and then to the brake oil pressure for deceleration management. The vertical control strategy diagram is illustrated in Figure 2.

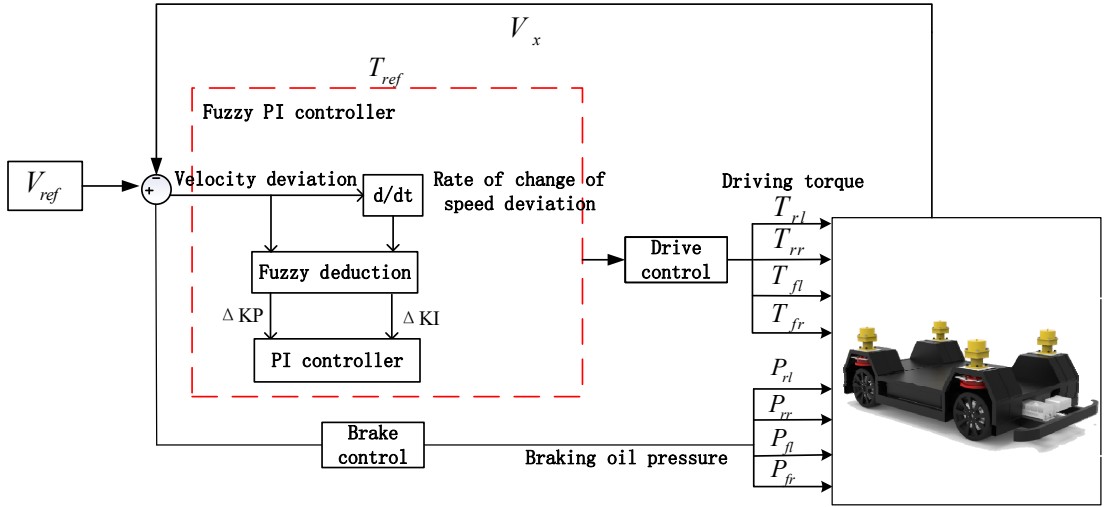

**Figure 2.** Vertical control strategy.

In Figure 2, $V_{ref}$ indicates the desired vehicle velocity, $V_x$ indicates the actual vehicle velocity, $T_{rl}$ indicates the drive moment at the left rear tire, $T_{rr}$ indicates the drive moment of the right rear tire, $T_{fr}$ indicates the drive moment of the right front tire and $T_{fl}$ indicates the drive moment of the left front tire. $P_{rl}$ indicates the brake oil pressure of the left rear tire, $P_{rr}$ indicates the brake oil pressure of the right rear tire, $P_{fr}$ indicates the brake oil pressure of the right front tire and $P_{fl}$ indicates the brake oil pressure of the left front tire.

### 2.1. Drive Control

The unmanned fully line-controlled distributed drive electric vehicle uses hub motors and only a conventional fuel vehicle model is available in the CarSim 2019 software. In order to build a drive system that is consistent with the properties of the real car motor, the motor is thus modelled externally. The motor model is built using PI control theory. The inputs to the motor model are the target drive torque and wheel velocity, and the actual drive torque is output through the regulation of the PI controller.

Fuzzy PI Controller Design

According to the controller input through the fuzzy controller online live-time regulation of the proportional coefficient (P), integral coefficient (I), the principle is illustrated in Figure 3.

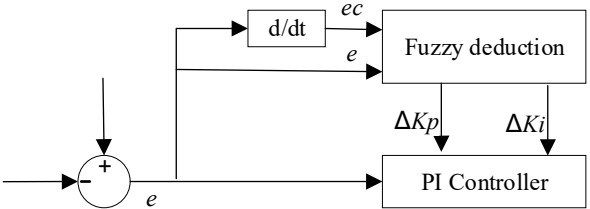

**Figure 3.** The principle of the fuzzy PI controller.

As in Figure 3, the controller's inputs are *e* (error of system) and *ec* (the rate of error change in the system). Fuzzy rules are formulated for reasoning decisions, and the two parameters of the PI controller are adjusted in live time with parameter changes of $\Delta K_p$ and $\Delta K_i$. The output is obtained by adding the changes to the initial values of the parameter values.

$$\begin{cases} K_p = K_{p0} + \Delta K_p \\ K_i = K_{i0} + \Delta K_i \end{cases} \tag{1}$$

In Equation (1), $K_{p0}$ and $K_{i0}$ are the initial values. $\Delta K_p$ and $\Delta K_i$ are the variation amounts rectified based on fuzzy control rules.

The *e* and *ec* are:

$$e = v_{ref} - v \tag{2}$$

$$ec = d/dt(e_v) \tag{3}$$

The range of speed deviation was formulated as $[-9, 9]$ and its theoretical domain range was set to $[-9, 9]$; the range of speed deviation rate was $[-3, 3]$ and its theoretical domain range was set to $[-3, 3]$; The theoretical domains of both $\Delta K_p$ and $\Delta K_i$ are $[-10, 10]$; the input and output quantities are used as triangular affiliation functions, as depicted in Figure 4.

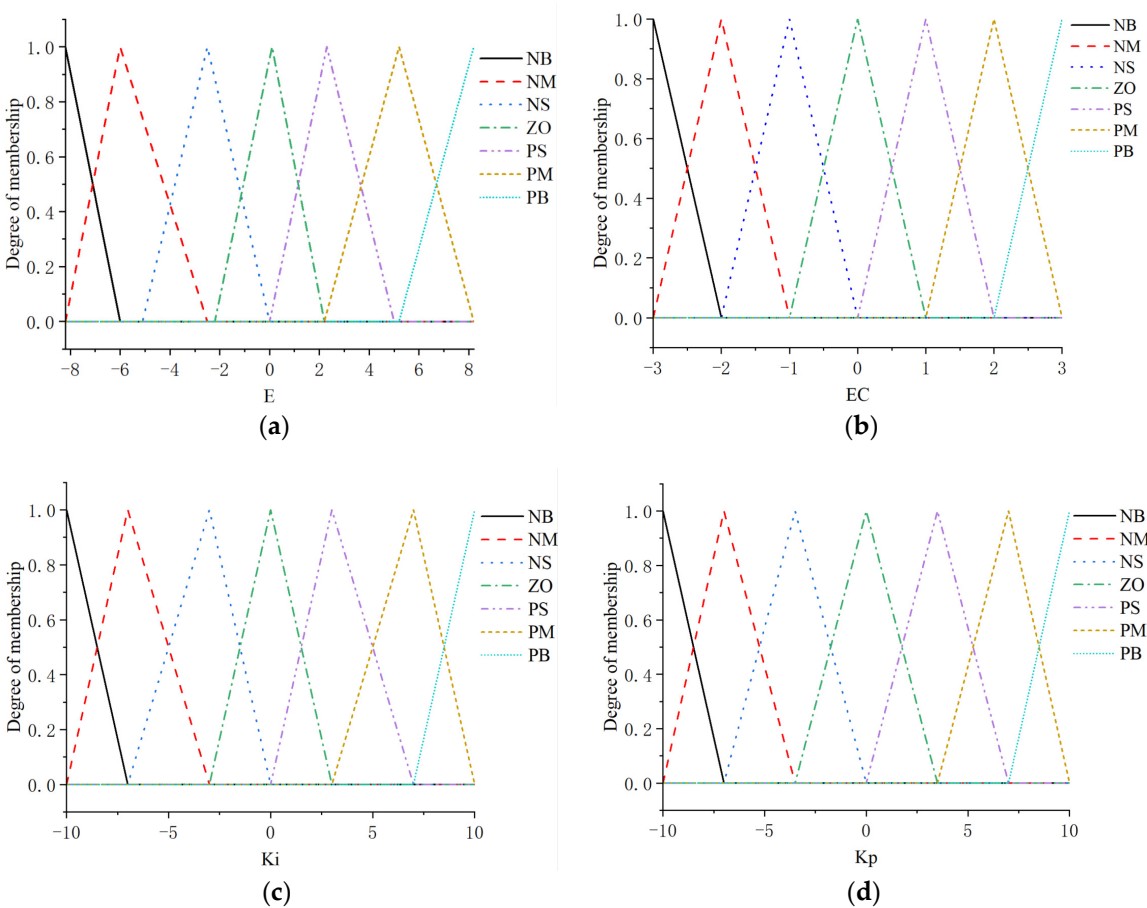

**Figure 4.** Input and output belong to the membership degree graph. (**a**) E membership degree; (**b**) EC membership degree; (**c**) $k_i$ membership degree; (**d**) $k_p$ membership degree.

As illustrated in Figure 4, the fuzzy set is divided into {negative large, negative average, negative small, zero, positive small, positive average, positive large}, i.e., {NL, NA, NS, ZO, PS, PA, PL}.

The vertical drive controller is mainly in accordance with the vague '*e*' and '*ec*' inference of the change values $K_p$ and $K_i$ the correction of the real-time PI data. When the vehicle is starting and stopping, the speed deviation is large, at which point the system

selects a somewhat larger value for $K_p$ in order to obtain a larger driving force to quickly eliminate errors; as E and the EC become medium, $K_p$ is reduced somewhat and a smaller $K_i$ is chosen; when the actual velocity is infinitely near the desired speed and the e is small, the output value is zero to prevent excessive speed control. The resulting fuzzy rules are indicated in Tables 1 and 2.

**Table 1.** Vague rules for $K_p$.

| $e$ | ec | | | | | | |
|----|----|----|----|----|----|----|----|
| | **NL** | **NA** | **NS** | **ZO** | **PS** | **PA** | **PL** |
| NL | PL | PL | PA | PS | NS | NA | NL |
| NA | PL | PL | PA | PS | NS | NA | NL |
| NS | PL | PA | PS | ZO | NS | NA | NL |
| ZO | PL | PA | PS | ZO | NS | NA | NL |
| PS | PL | PA | PS | ZO | NS | NA | NS |
| PA | PL | PS | PA | NS | NS | NA | NA |
| PL | PL | PS | PS | PS | NA | NA | NA |

**Table 2.** Vague rules for $K_i$.

| $e$ | ec | | | | | | |
|----|----|----|----|----|----|----|----|
| | **NL** | **NA** | **NS** | **ZO** | **PS** | **PA** | **PL** |
| NL | NL | NL | NA | NS | PS | PA | PL |
| NA | NL | NL | NA | NL | PS | PA | PL |
| NS | NL | NA | NS | ZO | PS | PA | PL |
| ZO | NA | NS | NA | ZO | PS | PA | PL |
| PS | NA | NS | NS | ZO | PA | PS | PA |
| PA | NS | NS | NS | PS | PA | PS | PA |
| PL | NS | NS | NS | NS | PA | PS | PA |

The fuzzy rule plane diagram of $K_p$, $K_i$ is depicted in Figure 5:

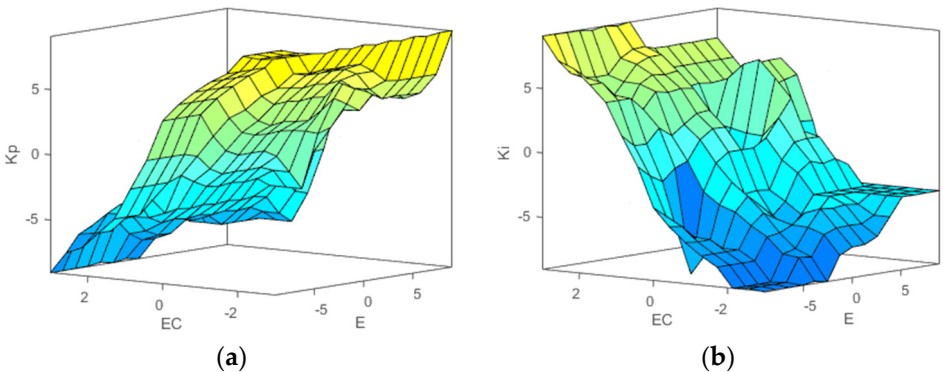

(**a**)   (**b**)

**Figure 5.** Fuzzy rule plane diagram of $K_p$, $K_i$. (**a**) Fuzzy rule plane diagram of $K_p$; (**b**) Fuzzy rule plane diagram of $K_i$.

*2.2. Brake Control*

Disc brakes have the benefits of being small in size, with good heat dissipation and high stability. Therefore, hydraulic disc brakes were selected as the braking system for an unmanned fully line-controlled distributed drive electric vehicle. The correlation between its wheel cylinder pressure and braking force is mainly through the wheel cylinder pressure acting on the brake friction pad, in the friction pad and the effective contact area of the brake

disc to produce a huge friction force, the vehicle kinetic energy into thermal energy process and finally the mechanical equation of the hydraulic disc brake obtained as Equation (4):

$$T_b = k \cdot r_b \cdot A \cdot P_c \tag{4}$$

where $T_b$ is the caliper torque; $k$ is the efficiency factor; $r_b$ is the equivalent friction radius; $A$ is the contact area; the operating pressure of the braking wheel cylinder is indicated by $P_c$.

When the current genuine velocity exceeds the target velocity, the automobile must be regulated to decelerate. The speed deviation is calculated to determine the vehicle's deceleration speed at this point in time, the required braking force is calculated using Newton's second law theory, and the braking pressure is distributed to the fore and aft axles in proportion to the stopping force of each wheel cylinder. Figure 6 depicts the brake control approach.

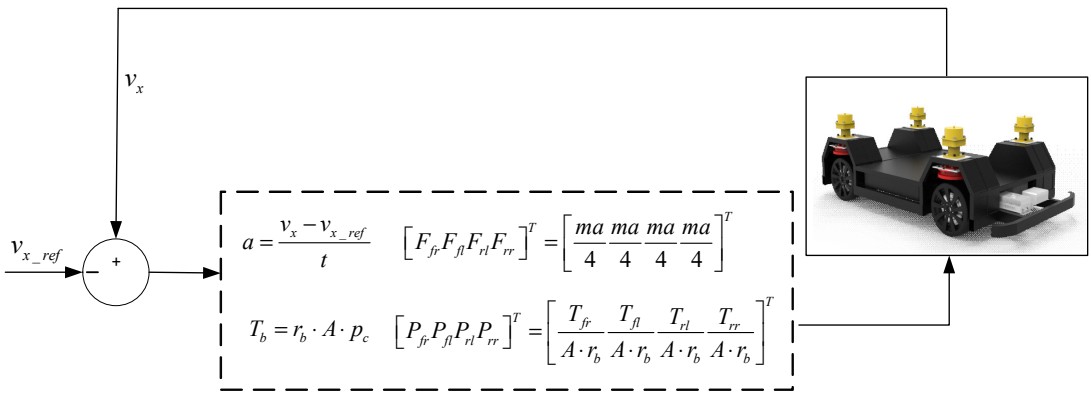

**Figure 6.** Brake control strategy diagram.

*2.3. Longitudinal Simulation Validation*

2.3.1. Overall Vehicle Parameters

A simulated vehicle model of an unmanned fully line-controlled distributed drive electric vehicle was built using CarSim 2019, with the dimensions of the car illustrated in Table 3.

**Table 3.** The car's parameters.

| Dimensions | Numerical Values | Unit |
|---|---|---|
| Car mass | 900 | kg |
| Range from center of mass to fore axis | 950 | mm |
| Range from center of mass to aft axis | 950 | mm |
| Car's width | 1680 | mm |
| Car's height | 1126 | mm |
| Wheelbase | 1400 | mm |
| Center of mass height | 450 | mm |
| Wheel radius | 287.6 | mm |
| $x$-axis rotational inertia | 280 | kg·m$^2$ |
| $y$-axis rotational inertia | 750 | kg·m$^2$ |
| $z$-axis rotational inertia | 750 | kg·m$^2$ |
| Spring load mass | 747 | kg |
| Gauge of the fore and aft axis | 1900 | mm |

2.3.2. Emulation Results Analysis

The whole vehicle model was built in CarSim 2019 to set three different road adhesion coefficients, high, medium and low, and was simulated jointly with Simulink to validate the longitudinal motion control algorithm. The emulation findings are shown in Figure 7.

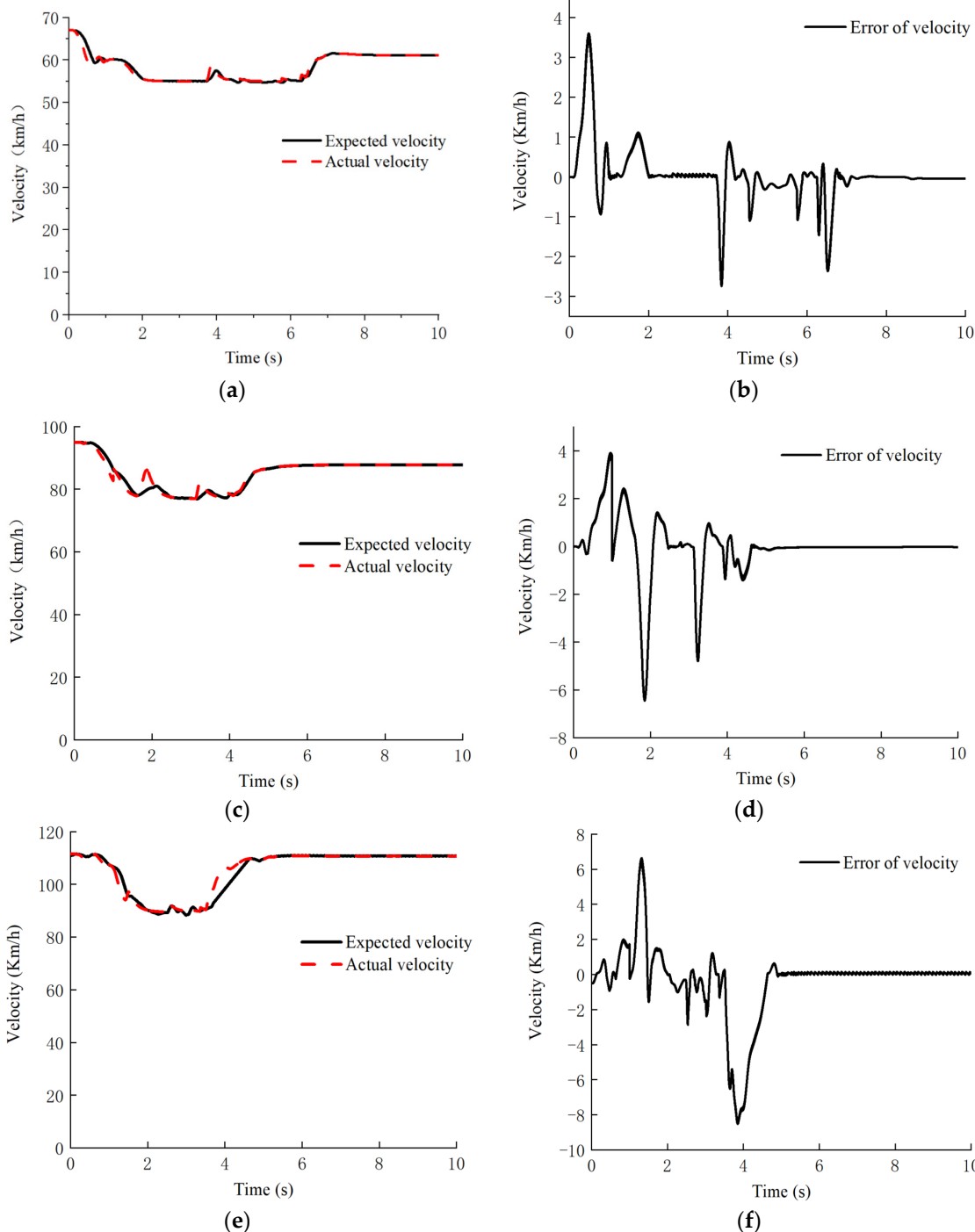

**Figure 7.** Emulation result. (**a,b**) shows the emulation of low adhesion coefficient pavement: pavement adhesion coefficient 0.3, starting velocity 67 km/h. (**a**) Comparison of actual speed and real speed; (**b**) Error of velocity; (**c,d**) shows the simulation of medium adhesion coefficient pavement: pavement adhesion coefficient 0.6, starting velocity 98 km/h; (**c**) Comparison of actual speed and real speed; (**d**) Error of velocity; (**e,f**) shows the simulation of higher adhesion coefficient pavement: 0.8, starting velocity 111.5 km/h; (**e**) Comparison of actual speed and real speed; (**f**) Error of velocity.

From Figure 7a,b, the actual velocity keeps track of the desired speed well, maximum speed error is within 4 km/h and its maximum error generation position is also in the just start, which indicates that the longitudinal motion controller keeps track of the desired speed well on roads with low traction. Figure 7c,d reveal that genuine velocity tracks the expected velocity, and the velocity error grows with velocity. At 2 s, there is a swing in the

vehicle velocity, at which time there is a large velocity error due to the fast velocity and the relatively small ultimate adhesion between the tires and the ground, and the maximum velocity error is within 7 km/h, indicating that the longitudinal motion controller can track the desired velocity well on the medium adhesion road. As described in Figure 7e,f, the actual velocity keeps track of the expected velocity. At 4 s, due to the fast velocity, the actual velocity exceeds the desired velocity at this time and needs to be decelerated. Considering that the high-speed emergency braking will cause vehicle stability, there is a large error in velocity tracking, but it exists for a short time and does not have a great impact on the overall effect. Therefore, on the high adhesion road, it shows that the longitudinal motion controller can track the desired velocity very well.

## 3. Lateral Control Strategy

IMU, GNSS and camera-based systems are commonly used to estimate the vehicle's states accurately. These systems provide various information, such as velocity, acceleration, sideslip angle and attitude, which are critical inputs for lateral control strategy prediction modeling. This article is based on the following references.

Reference [32] proposed a method for using IMU and GNSS data to evaluate the skid angle and posture of a car. The related work [33] proposed a method to evaluate the car's sideslip angle, considering the characteristics of the measurement signals. The method uses a combination of multiple sensors, including IMU and GNSS, and a neural network.

The lateral motion control of the unmanned fully line-controlled distributed drive electric vehicle includes lateral displacement control and front wheel angle control. According to its own sensing sensors, it collects information about the difference between its own status and the status of the reference path at the moment. It communicates the wheel turning angle to the steering system using MPC control algorithms to follow the intended path. The lateral control tactics are displayed in Figure 8.

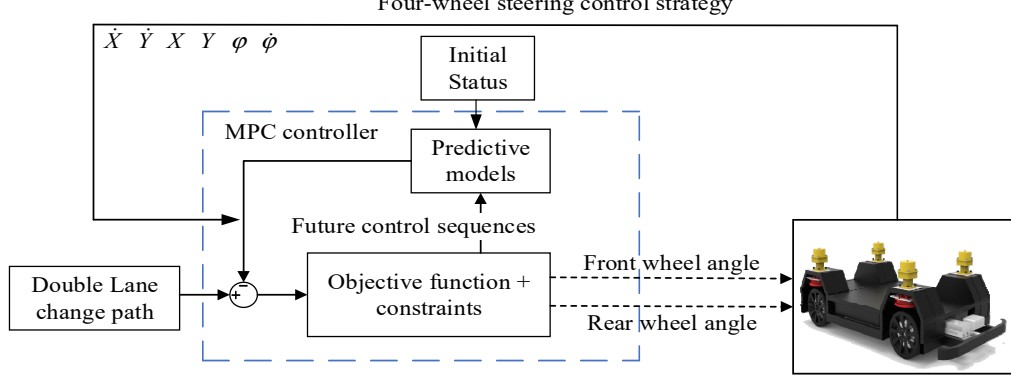

**Figure 8.** Lateral control strategy diagram.

As displayed in Figure 8, the fore and aft tires turning angles of the wheels are solved according MPC theory for prediction and optimization on the basis of an accurate model. The wheel angles calculated by the two controllers are fed to the vehicle's linear actuators to assure the vehicle follows the required trail precisely and stably. A joint simulation model is set up in the MATLAB/Simulink and CarSim 2019 software. Typical operating conditions are set up, and the simulation is validated on a high velocity and high adhesion coefficient and medium velocity and low adhesion coefficient road surface.

### 3.1. Lateral Control Strategy Prediction Modeling

The vehicle's dynamic control heavily relies on the vehicle dynamic model, which incorporates tire models such as that of Pacejka and Dugoff. Reference [34] proposes a method to estimate the sideslip angle of a smart car by integrating vehicle kinematics and dynamic Kalman filtering based on consensus, which improves the precision of the estimation.

The three-degrees-of-freedom monorail model was selected as the prediction model of the controller, as described in Figure 9.

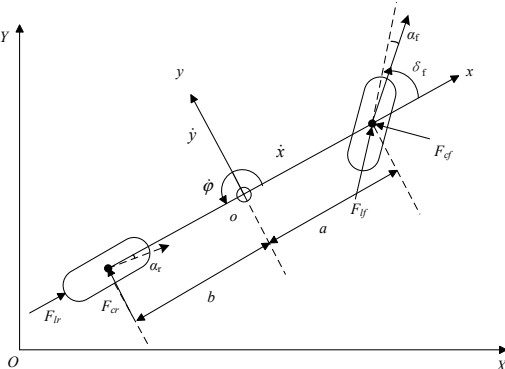

**Figure 9.** Three-degrees-of-freedom monorail model of vehicle.

By simplifying the tire model as well as using minor angles' assumptions, the forces on the tire are obtained as Equation (5):

$$\begin{cases} F_{lf} = C_{lf}S_f \\ F_{lr} = C_{lr}S_r \\ F_{cf} = C_{cf}(\delta_f - \frac{\dot{y}+a\dot{\varphi}}{\dot{x}}) \\ F_{cr} = C_{cr}(\frac{b\dot{\varphi}-\dot{y}}{\dot{x}}) \end{cases} \tag{5}$$

In Equation (5), $F_{lf}$ and $F_{lr}$ are the fore and aft tires vertical force; $C_{lf}$ and $C_{lr}$ are the fore and aft tires vertical stiffness; $S_f$, $S_r$ are the fore and aft tires slip rate; $F_{cf}$ and $F_{cr}$ are the fore and aft tires lateral force; $C_{cf}$ and $C_{cr}$ are the fore and aft tires lateral stiffness; $\delta_f$ is the vehicle fore and tire turning angle; $\dot{x}$ and $\dot{y}$ are the car velocity in the $x$ and $y$ axis orientation.

$$\begin{cases} \ddot{x} = \frac{2}{m}[C_{lf}S_f + C_{lr}S_r - C_{cf}(\delta_f - \frac{\dot{y}+a\dot{\varphi}}{\dot{x}})\delta_f - C_{cr}(\delta_r - \frac{\dot{y}-b\dot{\varphi}}{\dot{x}})\delta_f] + \dot{y}\dot{\varphi} \\ \ddot{y} = \frac{2}{m}[C_{cf}(\delta_f - \frac{\dot{y}+a\dot{\varphi}}{\dot{x}}) + C_{cr}(\delta_r - \frac{\dot{y}-b\dot{\varphi}}{\dot{x}})] - \dot{x}\dot{\varphi} \\ \ddot{\varphi} = \frac{2}{I_z}[aC_{cf}(\delta_f - \frac{\dot{y}+a\dot{\varphi}}{\dot{x}}) - bC_{cr}(\delta_r - \frac{\dot{y}-b\dot{\varphi}}{\dot{x}})] \\ \dot{Y} = \dot{x}\sin\varphi + \dot{y}\cos\varphi \\ \dot{X} = \dot{x}\cos\varphi - \dot{y}\sin\varphi \end{cases} \tag{6}$$

In Equation (6), $\ddot{x}$ is the increase in speed of the vehicle in the orientation of $x$; $\ddot{y}$ is the increase in speed of the vehicle in the orientation of $y$; $\dot{X}$ and $\dot{Y}$ are the velocity of the vehicle in the orientation of the $X$ and $Y$ axes of the geodetic coordinate system; $\dot{\varphi}$ is the yaw rate; $\delta_f$ is the angle of the fore wheel; $\delta_r$ is the angle of rotation of the aft wheel; $\alpha_f$ and $\alpha_r$ are the sideslip angles of the fore and aft wheels.

Rewrite Equation (6) as a state space expression, firstly, linearize the expression using Taylor series expansion; secondly, use forward Eulerian discretization for the constituted linear state space equation; finally, obtain the state space equation as Equation (7):

$$\dot{\xi}(t) = f(\xi(t), u(t)) \tag{7}$$

In Equation (7), $\xi(t)$ denotes the system state quantity, $u(t)$ denotes the system control quantity, $\xi(t) = [\dot{y}, \dot{x}, \varphi, \dot{\varphi}, Y, X]^T$, $u(t) = [\delta_f, \delta_r]$.

Linearizing Equation (7) at time t using Taylor expansion:

$$\dot{\xi}(t) = \dot{\bar{\xi}}(t) + A_t\xi(t) + B_tu(t) \tag{8}$$

In Equation (8), the Jacobi matrix $A_t = \frac{\partial f(\xi(t), u(t))}{\partial \xi}$, $B_t = \frac{\partial f(\xi(t), u(t))}{\partial u}$.

Equation (9) reveals the discrete linear time-varying system obtained by discretizing Equation (8) via the forward Euler formula:

$$\xi(k+1) = A_k\xi(k) + B_k u(k) \tag{9}$$

In Equation (9), $A_k = I + TA_t$, $B_k = TB_t$. The Jacobi matrix, $A_t$ and $B_t$, the calculation equations are as follows:

$$A(t) = \frac{\partial f(\xi(t), u(t))}{\partial \xi} = \begin{bmatrix} \frac{-2(C_{cf}+C_{cr})}{m\dot{x}_t} & \frac{\partial f_{\dot{y}}}{\partial \dot{x}} & 0 & -\dot{x}_t + \frac{2(bC_{cr}-aC_{cf})}{m\dot{x}_t} & 0 & 0 \\ \dot{\varphi} - \frac{2C_{cf}\delta_f}{m\dot{x}_t} & \frac{\partial f_{\dot{x}}}{\partial \dot{x}} & 0 & \dot{y}_t - \frac{2aC_{cf}\delta_f}{m\dot{x}_t} & 0 & 0 \\ 0 & 0 & 0 & 1 & 0 & 0 \\ \frac{2(bC_{cr}-aC_{cf})}{I_z\dot{x}_t} & \frac{\partial f_{\dot{\varphi}}}{\partial \dot{x}} & 0 & \frac{-2(a^2C_{cf}+b^2C_{cr})}{I_z\dot{x}_t} & 0 & 0 \\ \cos\varphi_t & \sin\varphi_t & \dot{x}_t\cos\varphi_t - \dot{y}\sin\varphi_t & 0 & 0 & 0 \\ -\sin\varphi_t & \cos\varphi_t & -\dot{y}_t\cos\varphi_t - \dot{x}_t\sin\varphi_t & 0 & 0 & 0 \end{bmatrix}$$

$$B_t = \frac{\partial f(\xi(t), u(t))}{\partial u} = \begin{bmatrix} \frac{C_{cf}}{m} & \frac{C_{cf}(\delta_f - \frac{\dot{y}_t + a\dot{\varphi}_t}{\dot{x}_t})}{m} & 0 & \frac{aC_{cf}}{I_z} & 0 & 0 \\ \frac{C_{cf}}{m} & \frac{C_{cf}(-\delta_r + \frac{\dot{y}_t - b\dot{\varphi}_t}{\dot{x}_t})}{m} & 0 & \frac{aC_{cf}}{I_z} & 0 & 0 \end{bmatrix}$$

$$\frac{\partial f_{\dot{y}}}{\partial \dot{x}} = \frac{\left[2C_{cf}(\dot{y} + a\dot{\varphi}) + 2C_{cr}(\dot{y}_t - b\dot{\varphi}_t)\right]}{m\dot{x}_t^2} - \dot{\varphi}_t, \quad \frac{\partial f_{\dot{\varphi}}}{\partial \dot{x}} = \frac{\left[2aC_{cf}(\dot{y} + a\dot{\varphi}) - 2bC_{cr}(\dot{y} - b\dot{\varphi})\right]}{I_z\dot{x}_t^2}$$

$$\frac{\partial f_{\dot{x}}}{\dot{x}} = \frac{C_{cf}\delta_f(\dot{y} + a\dot{\varphi})}{m\dot{x}_t^2} + \frac{C_{cf}\delta_r(\dot{y} - b\dot{\varphi})}{m\dot{x}_t^2}$$

### 3.2. Design Cost Function

The cost function is needed to determine the optimization of the system state quantities as well as the control quantities according to the cost function in the solution process. The cost function is shown in Equation (10):

$$J[\xi(t), u(t-1), \Delta U(t)] = \sum_{i=1}^{N_p} ||\eta(k+i\mid t) - \eta_{ref}(k+i\mid t)||_Q^2 + \sum_{i=1}^{N_c-1} ||\Delta u(k+i\mid t)||_R^2 + \rho\varepsilon^2 \tag{10}$$

In Equation (10), $\eta(k+i|t)$ is the practical outcome; $\eta_{ref}(k+i\big|t)$ is the reference outcome; $Q$ and $R$ are the weight matrices; the slack factor weight ratio is $\rho$; $\varepsilon$ is the slack ratio.

In cost function, the output at future moments needs to be predicted.

Convert Equation (9) as Equation (11):

$$\chi(k) = \begin{bmatrix} \xi(k) \\ u(k-1) \end{bmatrix} \tag{11}$$

A new expression for the status space equation is gained, as shown in Equation (12):

$$\begin{aligned} \chi(k+1) &= \widetilde{A}_k + \chi(k) + \widetilde{B}_k\Delta U(k) \\ \eta(k) &= \widetilde{C}_k\chi(k) \end{aligned} \tag{12}$$

In Equation (12), $\widetilde{A}_k = \begin{bmatrix} A_k & B_k \\ 0_{1*6} & I_1 \end{bmatrix}$, $\widetilde{B}_k = \begin{bmatrix} B_k \\ I_1 \end{bmatrix}$, $\widetilde{C}_k = \begin{bmatrix} C_k & 0_{2*1} \end{bmatrix}$, $C_k = \begin{bmatrix} 0 & 0 & 1 & 0 & 0 & 0 \\ 0 & 0 & 0 & 0 & 1 & 0 \end{bmatrix}$.

After the above derivation, the system prediction output expression is obtained as shown in Equation (13):

$$Y(t) = \psi\chi(t|t) + \Theta\Delta U(t) \tag{13}$$

$$\text{where } Y(t) = \begin{pmatrix} \eta(t+1\mid t) \\ \eta(t+2\mid t) \\ \cdots \\ \eta(t+N_p\mid t) \end{pmatrix}, \psi = \begin{pmatrix} \widetilde{C}_{t,t}\widetilde{A}_{t,t} \\ \widetilde{C}_{t,t}\widetilde{A}_{t,t}^2 \\ \cdots \\ \widetilde{C}_{t,t}\widetilde{A}_{t,t}^{N_p} \end{pmatrix}, \Delta U(t) = \begin{pmatrix} \Delta u(t\mid t) \\ \Delta u(t+1\mid t) \\ \cdots \\ \Delta u(t+N_c\mid t) \end{pmatrix}$$

$$\Theta = \begin{pmatrix} \widetilde{C}_{t,t}\widetilde{B}_{t,t} & 0 & 0 & 0 \\ \widetilde{C}_{t,t}\widetilde{A}_{t,t}\widetilde{B}_{t,t} & \widetilde{C}_{t,t}\widetilde{B}_{t,t} & 0 & 0 \\ \widetilde{C}_{t,t}\widetilde{A}_{t,t}^2\widetilde{B}_{t,t} & \widetilde{C}_{t,t}\widetilde{A}_{t,t}\widetilde{B}_{t,t} & \widetilde{C}_{t,t}\widetilde{B}_{t,t} & 0 \\ \cdots & \widetilde{C}_{t,t}\widetilde{A}_{t,t}^2\widetilde{B}_{t,t} & \widetilde{C}_{t,t}\widetilde{A}_{t,t}\widetilde{B}_{t,t} & \widetilde{C}_{t,t}\widetilde{B}_{t,t} \\ \widetilde{C}_{t,t}\widetilde{A}_{t,t}^{N_p-2}\widetilde{B}_{t,t} & \cdots & \cdots & \cdots \\ \widetilde{C}_{t,t}\widetilde{A}_{t,t}^{N_p-1}\widetilde{B}_{t,t} & \widetilde{C}_{t,t}\widetilde{A}_{t,t}^{N_p-2}\widetilde{B}_{t,t} & \cdots & \widetilde{C}_{t,t}\widetilde{A}_{t,t}^{N_p-N_c-1}\widetilde{B}_{t,t} \end{pmatrix}$$

To facilitate computer solutions, the cost function needs to be transformed into a simple quadratic shape, as shown in Equation (14):

$$J(\xi(t), u(t-1), \Delta U(t)) = [\Delta U(t)^T, \varepsilon]^T H_t[\Delta U(t)^T, \varepsilon] + G_t[\Delta U(t)^T, \varepsilon] \tag{14}$$

In Equation (14), $H_t = \begin{bmatrix} \Theta_t^T Q\Theta_t + R & 0 \\ 0 & \rho \end{bmatrix}$, $G_t = \begin{bmatrix} 2e_t^T Q\Theta_t & 0 \end{bmatrix}$.

The optimized control increase acting on the system is derived by tackling the problem in conjunction with the following constrained optimization problem:

$$\begin{cases} \Delta U_{\min} \leq \Delta U_t \leq \Delta U_{\max} \\ U_{\min} \leq U_t \leq U_{\max} \\ y_{\text{hc,min}} \leq y_{\text{hc}} \leq y_{\text{hc,max}} \\ y_{\text{sc,min}} - \varepsilon \leq y_{\text{sc}} \leq y_{\text{sc,max}} + \varepsilon \end{cases} \tag{15}$$

In Equation (15), $y_{\text{hc}}$ and $y_{\text{sc}}$ are the hard and soft constraint outputs, respectively; $\varepsilon$ is the relaxation factor.

Equation (15) is solved to obtain the optimum order of increasing front wheel rotation angle as in Equation (16):

$$\Delta U_t^* = (\Delta u, \Delta u_{t+1}^*, \cdots, \Delta u_{t+N_c-1}^*, \varepsilon)^T \tag{16}$$

The initial term of the increasing series is applied as the control increment incoming to the system, which gives Equation (17):

$$u(t) = u(t-1) + \Delta u_t^* \tag{17}$$

Go to the following sample moment and duplicate the mentioned calculation process to enable control of the situation.

### 3.3. Establishing Constraints

In the case of avoidance, the controller, in the process of solving the step signal, causes the vehicle to be in the trajectory tracking driving process of dangerous conditions; this needs certain constraints to be added on the vehicle state, such as Equation (18):

$$\begin{aligned} -25° \leq \delta_f \leq 25° \\ -5° \leq \delta_r \leq 5° \\ -0.85° \leq \Delta\delta_f \leq 0.85° \end{aligned} \tag{18}$$

### 3.4. Emulation Verification

The lateral motion strategy model is established in MATLAB/Simulink, the simulated automotive model is established in CarSim 2019, the high attachment coefficient of 0.85 and the low attachment coefficient of 0.3 are set for typical conditions to simulate the trajectory tracking lateral control, and the results are shown in Figures 10 and 11.

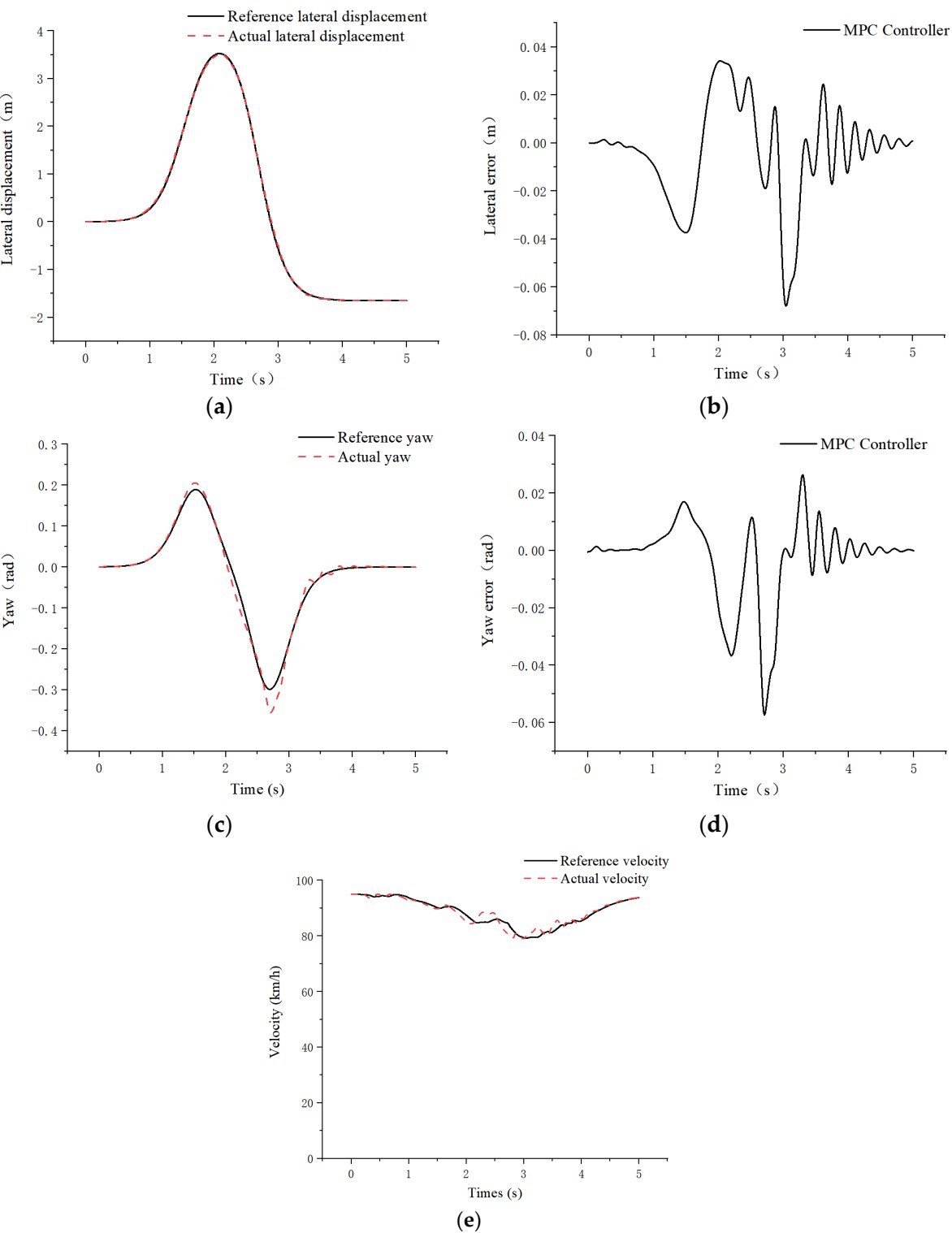

**Figure 10.** Emulation outcomes of high attachment conditions. (**a**) Chart of reference lateral displacement and actual lateral displacement; (**b**) Lateral error diagram; (**c**) Comparison of the reference yaw and the actual yaw; (**d**) Yaw error diagram; (**e**) Reference velocity and actual velocity comparison chart.

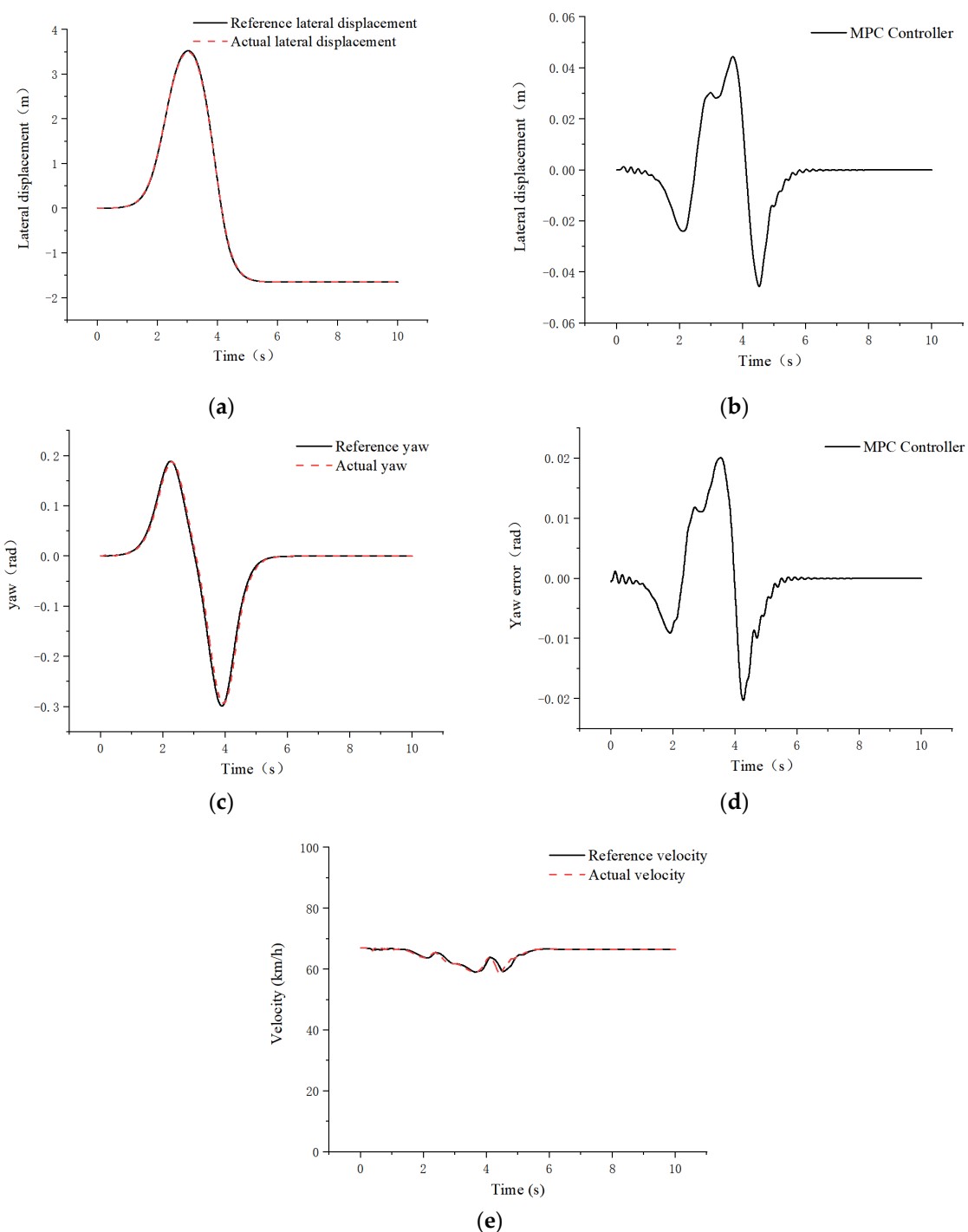

**Figure 11.** Emulation result of low adhesion coefficient.(**a**) Chart of reference lateral displacement and actual lateral displacement; (**b**) Lateral error diagram; (**c**) Comparison of the reference yaw and the actual yaw; (**d**) Yaw error diagram; (**e**) Reference velocity and actual velocity comparison chart.

### 3.4.1. High-Speed High Adhesion Coefficient Condition

The trajectory tracking was performed on a pavement surface with an adhesion coefficient μ of 0.85. Figure 10 exhibits the emulation results.

Figure 10a,c,e illustrate how the MPC controller could follow the target path with the greatest transverse fault of 0.068 m in Figure 10b. From Figure 10d, it is evident that the yaw angle deviation is 0.059 rad. The MPC controller offers good trajectory following control at high speeds and on roads with high adhesion.

### 3.4.2. Medium Speed Low Adhesion Coefficient Condition

The trajectory tracking was performed on a pavement surface with an adhesion coefficient μ of 0.3, and the emulation outcomes are presented in Figure 11.

From Figure 11a,c,e, it is evident that the MPC controller could well follow the target way, with a greatest transverse fault of 0.046 m as illustrated in Figure 11b. Figure 11d shows the largest yaw angle variance of 0.02 rad, all less than 0.1. The suggested track-following control policy performs better on the low adhesion coefficient pavement.

## 4. Fault-Tolerance Control Strategy

The fault-tolerant control technique is designed to ensure the safe driving of unmanned fully line-controlled distributed drive electric vehicles when the steering motor fails during track tracking. Figure 12 depicts the line-controlled steering control strategy diagram.

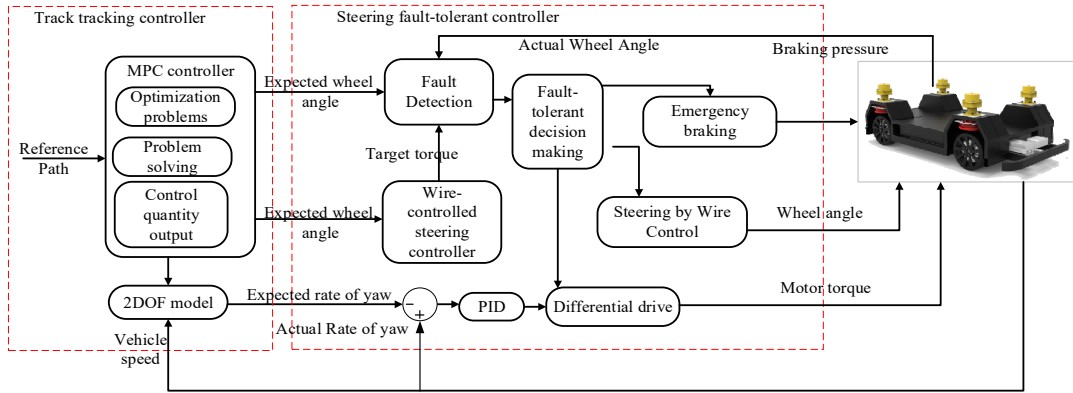

**Figure 12.** Fault-tolerance control strategy.

As displayed in Figure 12, the fault detection module monitors the status of the line-controlled steering system of an unmanned fully line-controlled distributed drive electric vehicle in real time.

1.  When there is no fault in the steering by wire, the vehicle is driven normally, and the cornering angle value of the vehicle is solved by the track following controller, and the solved steering angle is input to the steering controller established in MAT-LAB/Simulink. The steering controller transmits the target torque to the steering motor through calculation, and then the steering motor is input to the simulated vehicle model in CarSim 2019.
2.  When there is a fault in the steering by wired control, the fault detection module transmits the number and position information of the faulty motor to the fault-tolerant decision module. If the single motor fails, the fault-tolerant decision will control the differential drive block to control the car's yaw torque, so that the unmanned fully line-controlled distributed drive electric vehicle can maintain stability during the track following process until the vehicle stops. If there is multi-motor failure, the vehicle's steering-by-line system will fail more seriously, which will cause a particularly serious accident if the trajectory tracking continues, so the vehicle will then need to execute emergency braking to make the vehicle stop in the shortest time to ensure the vehicle is secure.

### 4.1. Fault Detection

The fault detection module detects the vehicle's steering system in real time during the vehicle track tracking process and detects the status information of the faulty motor during operation, mainly the target angle, actual angle, goal and practical moments' information of the motor. The main status information is the target angle, actual angle, goal and practical moments, etc. The steering system failure is determined by judging whether the ratio of

actual angle to target angle and the ratio of actual torque to target torque are greater than a fixed value.

Introducing the fault diagnosis factor λ as Equation (19):

$$\lambda_i = \frac{\delta_{\text{Actual}}}{\delta_{\text{Target}}}, \ \lambda_j = \frac{T_{\text{Actual}}}{T_{\text{Target}}} \tag{19}$$

In Equation (19), $\lambda_i$ is the angle fault factor and $\lambda_j$ is the torque fault factor.

Considering the relationship between the vehicle's own load and the road adhesion factor, the fault factor is selected as 0.85. Therefore, by determining whether the fault factor thresholds for both the angle and torque are less than 0.85, if they are less than this value, the steering system is considered to have a fault. The fault detection strategy is described in Figure 13.

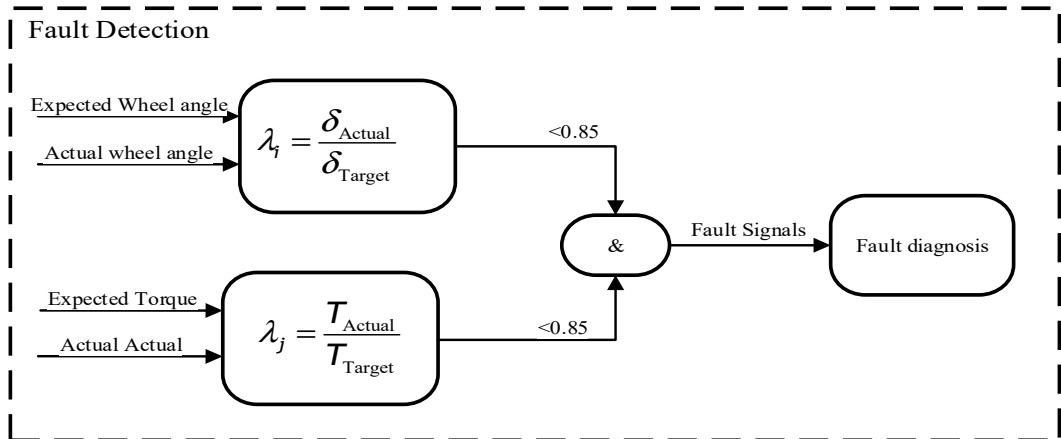

**Figure 13.** Fault detection strategy diagram.

### 4.2. Fault-Tolerant Decision Making

Based on the resulting steering-motor failure information, a decision is made about the next moment of vehicle response based on the location and number of steering motor failures. If all four steering motors fail at the same time, the vehicle is uncontrollable at that point and needs to stop quickly to enter the emergency braking state. When one or more steering motors fail, they can continue to work in some cases. Therefore, the failure of the motors is analyzed, as shown in Table 4.

**Table 4.** Fault-tolerant decision making.

| | Left Front Steering Motor | Right Front Steering Motor | Left Rear Steering Motor | Right Rear Steering Motor | Decision-Making Results |
|---|---|---|---|---|---|
| Single motor failure | ⊗ | √ | √ | √ | Fault tolerance control |
| | √ | ⊗ | √ | √ | Fault tolerance control |
| | √ | √ | ⊗ | √ | Fault tolerance control |
| | √ | √ | √ | ⊗ | Fault tolerance control |
| Dual motors failure | ⊗ | ⊗ | √ | √ | Rear wheel steering |
| | ⊗ | √ | ⊗ | √ | Emergency brake |
| | ⊗ | √ | √ | ⊗ | Emergency brake |
| | √ | √ | ⊗ | ⊗ | Front wheel steering |
| | √ | ⊗ | ⊗ | √ | Emergency brake |
| | √ | ⊗ | √ | ⊗ | Emergency brake |
| Triple motors failure | ⊗ | ⊗ | ⊗ | √ | Emergency brake |
| | ⊗ | ⊗ | √ | ⊗ | Emergency brake |
| | ⊗ | √ | ⊗ | ⊗ | Emergency brake |
| | √ | ⊗ | ⊗ | ⊗ | Emergency brake |
| Four motors failure | ⊗ | ⊗ | ⊗ | ⊗ | Emergency brake |

As shown in Table 4, ⊗ for a failed motor condition, √ for a normal motor operation. Faults can be classified as single motor faults, dual motors faults, triple motors faults and four motors faults.

(1)  Single motor failure: When a single steering motor failure occurs, differential drive control is performed.

(2)  Dual motors failure:

1.  Two front motors' failure: When the front two steering motors fail, the faulty motor stops working and the vehicle continues to drive through the rear wheel steering.

2.  Two rear motors' failure: When the two steering motors at the rear end fail, the rear axle steering motor can be made to stop working and drive using the front wheel steering form.

3.  Same side or opposite side motor failure: To ensure the security of the vehicle, use the emergency brake command to bring the vehicle to a quick stop.

(3)  Triple motors' failure: To ensure the security of the vehicle, use the emergency brake command to make the vehicle stop quickly.

(4)  Four motors' failure: To ensure the security of the vehicle, use the emergency brake command to make the vehicle stop quickly.

### 4.3. Differential Drive Control

When the failure of a single steering motor occurs during trajectory tracking of an unmanned fully line-controlled distributed drive electric vehicle, the vehicle is controlled in a fault-tolerant manner by using differential drive. The main working principle of differential drive is shown in Figure 14.

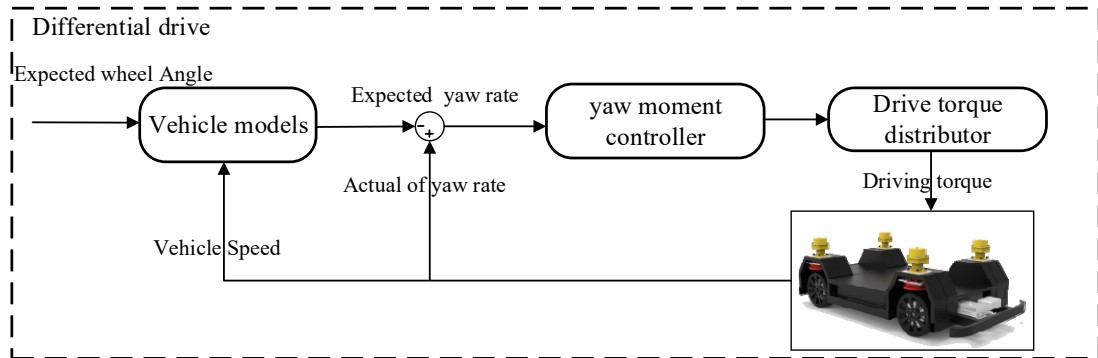

**Figure 14.** Working principle of differential drive.

As demonstrated in Figure 14, the trajectory tracking-control strategy resolves the expected wheel angle and allows the desired transverse swing angular velocity to be solved. The yaw torque controller selects the PID control technique, which receives the yaw angular velocity difference as input and outputs excess transverse swing torque, which is distributed to the separate wheel drive motors via the drive force distributor.

#### 4.3.1. Reference Model

The reference model was a simple two-degrees-of-freedom auto dynamic model, as demonstrated in Figure 15.

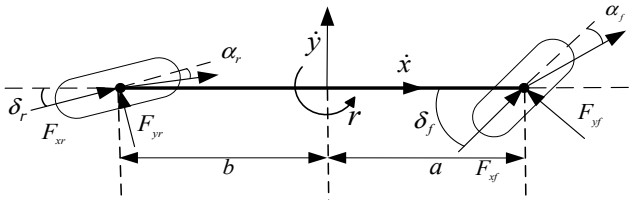

**Figure 15.** Linear two-degrees-of freedom model.

Based on the forces, the differential equation of motion for the vehicle are given as follows:

$$(k_1 + k_2)\beta + \frac{1}{u}(ak_1 - bk_2)w_r - k_1\delta_f - k_2\delta_r = m(\dot{v} + uw_r) \tag{20}$$

$$(ak_1 - bk_2)\beta + \frac{1}{u}(a^2k_1 + b^2k_2)w_r - ak_1\delta_f + bk_2\delta_r = I\dot{w}_r \tag{21}$$

where $m$ denotes the vehicle mass; $\delta_f$, $\delta_r$ denote the fore and aft wheel angle, respectively; $v$ denotes the lateral velocity; $u$ denotes the vertical speed; $w_r$ denotes the yaw; $\beta$ is the center of mass's side slip angle; $k_1$, $k_2$ the cornering stiffness of the fore and aft.

The rate of yaw is chosen as the control parameter of the steady-state response, and the rate of yaw in Equation (22):

$$w_{rd} = \frac{u/L}{1 + Ku^2}\delta_f \tag{22}$$

where $K$ is the stability factor.

$$K = \frac{m}{L^2}\left(\frac{a}{k_2} - \frac{b}{k_1}\right) \tag{23}$$

At the same time, considering the impact of vehicles on the road in the process of driving, its critical value is as in Equation (24):

$$w_{rd\_bound} = 0.85\frac{\mu g}{u} \tag{24}$$

### 4.3.2. Yaw Moment Controller

The PID controller, which is commonly used in industry, is selected for the yaw controller. The swing torque controller receives as input the difference between the ideal and intended yaw rates, and the output is the additional yaw torque required to stabilize the vehicle.

$$\Delta T = K_P e(t) + K_I \int e(t)dt + K_D \frac{de(t)}{dt} \tag{25}$$

In Equation (25), $K_P$ is the ratio element, $K_I$ is the integral element, $K_D$ is the differential element, $\Delta T$ is the additional yaw moment, and $e(t)$ is the yaw rate difference.

Figure 16 depicts the yaw torque controller's structure:

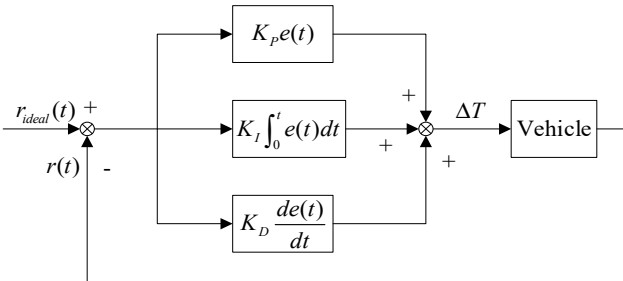

**Figure 16.** Yaw moment controller.

After several simulation experiments to adjust the parameters, our PID scale coefficients are $K_P = 100$, $K_I = 0.001$, $K_D = 0.1$.

### 4.3.3. Drive Force Divider

The drive force distributor is a yaw torque controller solution that distributes the drive motor torque to each drive wheel in a certain way. In this section, the yaw moment rules based on the motor characteristics are assigned. The additional yaw moment is indirectly converted into driving force by increasing the driving force of the drive motor on one side and decreasing the drive motor on the other side of the drive.

In the absence of steering motor failure, the torque of the car drive is distributed equally among the four wheels in the unmanned fully line-controlled distributed drive electric vehicle, and the total drive force is shown in Equation (26):

$$T = \left( F_{fl} + F_{fr} + F_{rl} + F_{rr} \right) \times r \tag{26}$$

In Equation (26), $T$ stands for the overall driving moment; $F_{fl}$, $F_{fr}$, $F_{rl}$ and $F_{rr}$ are the vertical forces acting on the left front wheel, right front wheel, left rear wheel, and right rear wheel, respectively; and r is the tire's rolling radius.

Equation (27) depicts the extra yaw moments:

$$M_z = \left( F_{fl} - F_{fr} + F_{rl} - F_{rr} \right) \times \frac{B}{2} \tag{27}$$

$M_z$ stands for the extra yaw moment in Equation (27), and $B$ stands for the wheelbase.

The peak torque of the car's drive motor is also bound to be limited by Equation (28) because the pavement's adhesion coefficient affects the longitudinal drive of the automobile:

$$- \mu F_z r \le T \le \min(\mu F_z r, \ T_{\max}) \tag{28}$$

where $T_{\max}$ is the peak moment of the drive motor.

When an additional yaw moment is imposed, the vehicle has two main states of steering motion, one is left turn understeer and the other is right turn oversteer. At this time, in order for the vehicle to be able to drive stably, the motion of the vehicle needs to be corrected by appropriately increasing the right-hand wheel drive torque and reducing the left-hand wheel drive torque. At this time, the torque distribution of each wheel is as Equations (29) and (30):

$$T_{fl,rl} = \frac{T}{4} - 0.25|M_z| / \frac{B}{2} \times r \tag{29}$$

$$T_{fr,rr} = \frac{T}{4} + 0.25|M_z| / \frac{B}{2} \times r \tag{30}$$

When the additional transverse moment $M_z < 0$, the vehicle mainly has two steering motion states, one is right turn understeer and the other is left turn oversteer. At this time, in order for the vehicle to be able to drive stably, the motion status of the vehicle needs to be corrected by appropriately increasing the drive moment for the left wheel and reducing the drive moment for the right wheel. The torque distribution of each wheel is as follows:

$$T_{fl,rl} = \frac{T}{4} + 0.25|M_z| / \frac{B}{2} \times r \tag{31}$$

$$T_{fr,rr} = \frac{T}{4} - 0.25|M_z| / \frac{B}{2} \times r \tag{32}$$

### 4.4. Simulation Verification

The simulated automobile model is built up in CarSim 2019 for various failure mode simulation experiments, and the fault-tolerant control tactics model is developed in MATLAB/Simulink. The simulation verification is shown as follows.

### 4.4.1. Single Motor Fault Conditions

When the left steering motor on the tracked vehicle breaks after 2 s of normal operation, the feasibility of the fault-tolerant algorithm is confirmed by contrasting the two

approaches for both fault-tolerant control and fault-tolerant control alone. Figure 17 details the emulation outcomes.

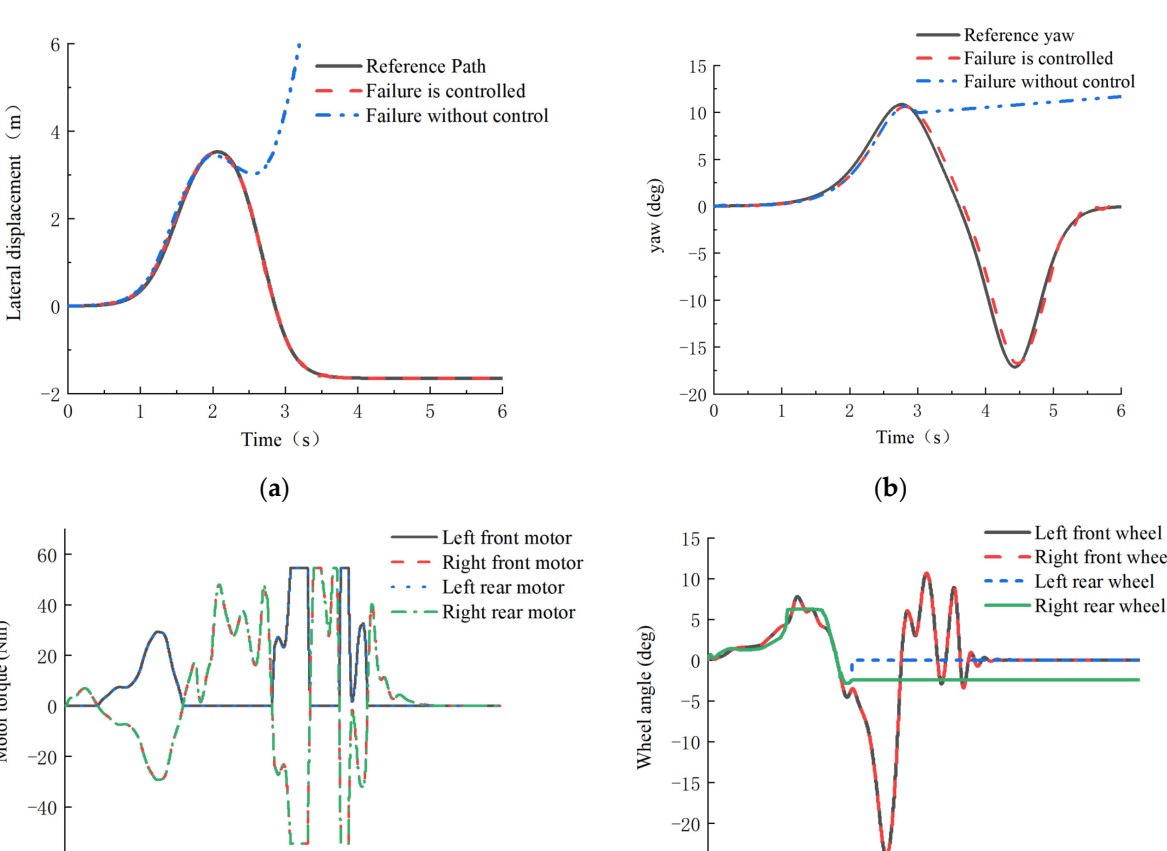

**Figure 17.** Single motor fault simulation results. (**a**) Graph of transverse displacement; (**b**) Graph of yaw angle; (**c**) Motor torque diagram; (**d**) Wheel turning angle comparison chart.

As can be seen in Figure 17, a left rear motor failure occurs during the vehicle's track tracking process, at which point the vehicle requires a differential drive operation command. Before 2 s, the vehicle can accurately trail the double lane path. It is obvious that the vehicle lacks fault-tolerant control at this point since the left steering motor has a failed turn angle value of 0 degrees at 2 s. Since the motor failure will cause the vehicle to understeer, the vehicle will find it difficult to follow the course and flee. However, a vehicle with fault-tolerant regulation is able to accurately continue to track the path. The proposed fault-tolerant control regulation for steering by wire can improve the safety and stabilization of the vehicle when a single motor fails.

### 4.4.2. Three Motors Fault Working Conditions

The vehicle was tracked in typical operating conditions, and at 2 s, the triple steering motors failed, at which point it went into emergency braking.

From Figure 18, it can be seen that in 2 s when there is vehicle right front, right rear, left rear wire-controlled steering motor failure, at this moment in time the vehicle, according to the fault-tolerant decision-making instructions for emergency braking, quickly reaches the maximum 10 MPa in the four wheels of the wheel cylinder pressure, as shown in Figure 18c; in Figure 18d, the vehicle speed in the implementation of emergency braking can be seen. The vehicle speed slows down quickly in 2 s from 85 km/h down to 0 km/h, so that the vehicle stops in the shortest possible time to ensure the security of the vehicle.

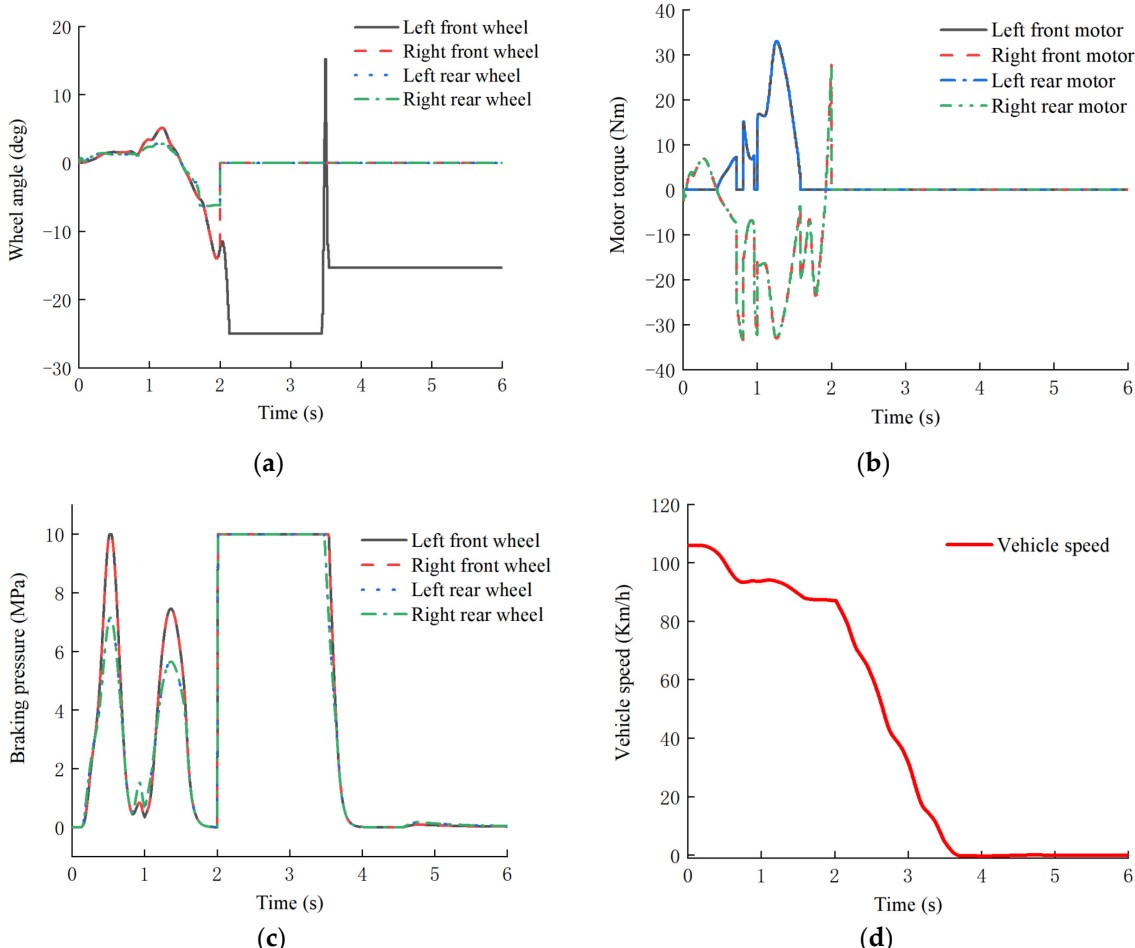

**Figure 18.** Three motor fault simulation results. (**a**) Vehicle wheel angle diagram; (**b**) Motor torque diagram; (**c**) Brake pressure diagram; (**d**) Vehicle velocity diagram.

## 5. Conclusions

In this article, the unmanned fully line-controlled distributed drive electric vehicle is the main object of study, takes the dynamic model as the basis, designs the vehicle's horizontal and vertical motion control strategy and improves the trajectory tracking accuracy as much as possible while ensuring the stability. Meanwhile, considering the characteristics of the unmanned fully line-controlled distributed drive electric vehicle with multiple sensors and multiple actuators will have the risk of failure, the wire-controlled steering fault-tolerant control tactics are designed to enhance the safety of the vehicle. Through simulation experiments, the in-line steering fault-tolerant control approach as well as the vertical and horizontal motion control strategies for the vehicle are also examined. This page provides some useful references for the path-following control of distributed-drive, completely autonomous electric cars.

1.  Systematic path tracking and fault-tolerant control of an unmanned fully line-controlled distributed drive electric vehicle. Motor failure can cause instability in the tracking of paths.
2.  In the trajectory tracking-control study, the fuzzy PI control strategy is used for longitudinal control, and the MPC strategy is used for horizontal control. The largest lateral mistake is 0.068 m and the biggest yaw angle mistake is 0.059 rad in high-speed turning simulations under high and low road adhesion coefficients. The track-following control policy is precise and stable.
3.  Differential drive technique with yaw momentum and torque distribute controllers provide fault-tolerant control. The results reveal that when a steering motor fails, the fault-tolerant control technique can maintain trajectory tracking safety; when three

steering motors fail, the vehicle can operate the emergency brake command, and the vehicle can quickly stop within 2 s to secure the vehicle during the track following.

At the same time, this study also has some limitations. In the current study, the adaptive hierarchical trajectory tracking-control method has not been validated on real vehicles. Therefore, future research can further explore the feasibility and validity of the method in practical application scenarios. To properly validate the strategy, experimental data from real automobiles are compared to simulations.

Given that the lateral slew angle of the vehicle in this paper is difficult to collect by sensors in practice, most current solutions are obtained by the estimation of algorithms, which has a certain error, and subsequent research must consider how to increase the precision of the mass's lateral slew angle.

**Author Contributions:** Conceptualization, T.T., G.L. and Y.L.; methodology, T.T.; software, Y.L., H.B.; validation, T.T., G.L. and N.L.; formal analysis, Y.L.; investigation, T.T.; resources, G.L.; data curation, H.B.; writing—original draft preparation, T.T.; writing—review and editing, T.T. and G.L.; visualization, T.T. and N.L. supervision, G.L.; project administration, G.L.; funding acquisition, G.L. All authors have read and agreed to the published version of the manuscript.

**Funding:** This work was supported by the Liaoning Provincial Natural Fund Grant Program Project (2022-MS-376).

**Institutional Review Board Statement:** Not applicable.

**Informed Consent Statement:** Not applicable.

**Data Availability Statement:** The data used to support the findings of this study are available from the corresponding author upon request.

**Conflicts of Interest:** The authors declare no conflict of interest.

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
