# Peer review of "Trajectory Tracking Control Study of Unmanned Fully Line-Controlled Distributed Drive Electric Vehicles"

_applsci, doi:10.3390/app13116465_

Round 1

Reviewer 1 Report

The article is quite professionally written, touches on current topics.

Key notes:

- a description of the results obtained should be added to the annotation;

- fix minor design errors (lines 138, 405);

- in the conclusions, one should compare the numerical characteristics obtained during the simulations.

Minor editing of English language required

Reviewer 2 Report

A brief summary

In this study, a layer based trajectory tracking control strategy is proposed and validated using simulations. Author used fuzzy rules for longitudinal control and model based technique for later control. Further, various fault tolerant control is also presented. All the developed control techniques are validated using CarSim software.

General concept comments

1.      The author did not highlight the novelty of this work. I recommend the author to strongly state the novelty of the paper in both abstract and in introduction. Also, compare the performance of proposed control technique with state of the art. 

2.      Overall figures need to be improved. Font size, legend, and line style should be clear.

3.      Some of the control theory and model development shall be replaced by citing appropriate work. This reduces the overall length of the paper.

Specific comments

1.      Line 124: Section 3, title is wrong. Please update.

2.      Figure 4: Insert legend and change line style in the same plot.

3.      Line 247: Based on figure, Velocity should be around 110 km/h

Quality of English is good. 

Reviewer 3 Report

This paper presents a novel method for vehicle trajectory tracking control. Overall, the research is potential and the result is convincing. Some minor modifications should be made before publication.

1) For the lateral control strategy prediction modelling, how to obtain accurate vehicle states is challenging. Currently, some work based on multi-sensor fusion such as GNSS, IMU, and cameras are deployed to estimate vehicle states such as velocity, sideslip angle, acceleration, and attitude precisely. Thus, the following related work should be included: imu-based automated vehicle body sideslip angle and attitude estimation aided by gnss using parallel adaptive kalman filters, automated vehicle sideslip angle estimation considering signal measurement characteristic, estimation on imu yaw misalignment by fusing information of automotive onboard sensors.

2) As intelligent vehicles continue to evolve, electric vehicles are being equipped with an increasing number of sensors. Nonetheless, ensuring that different sensors are synchronized in terms of time and space is crucial for accurate vehicle control signal input. Currently, there is ongoing research focusing on this topic: yolov5-tassel: detecting tassels in rgb uav imagery with improved yolov5 based on transfer learning. automated driving systems data acquisition and processing platform. It would be better to discuss the above work in the introduction.

3) The vehicle's dynamic control heavily relies on the vehicle dynamic model, which incorporates tire models such as Pacejka and Dugoff. Some related work should be included as follows: autonomous vehicle kinematics and dynamics synthesis for sideslip angle estimation based on consensus kalman filter, improved vehicle localization using on-board sensors and vehicle lateral velocity.

4) Figure 1 should be optimized as the distribution layer also belongs to executive layer in my opinion. Furthermore, the quality of Figure 4 should also be improved.

5) There are some Chinese words in Figure 13. Please modify them. In addition, please add the work limitation and future work in the conclusion.

6)  Please explain in detail why some control modules are based on PID, and the lateral control module is based on MPC.

Reviewer 4 Report

The manuscript presents an original scientific achievement in the form of the synthesis of control of braking, driving, steering and fault-tolerant system of electric vehicle when some actuators fail to ensure the stability of trajectory tracking accuracy and safety of the vehicle.

As it stands, the manuscript is too much of a research report than a scientific paper.

Comments:

1. The thesis of the paper, presented not very clearly in Section 2, should be moved to the final part of Section 1.

2. Section 3 should probably be titled "Longitudinal control strategy".

3. Section 5.3.2. Yaw momentum controller, an ideal PID controller is shown in Figure 16, but the actual controller used is not specified, especially the D component, to complete the method of selecting the optimal setting values of this controller.

4. In line 540, "In equation 34", I think it should be "In equation 27".

5. Section Conclusion should be numbered 6.

6. Complete the Section Conclusion with a detailed plan for further research on the paper.

7. Generally, most drawings are hard to read.

8. Try to shorten the text of the manuscript a bit, leaving those fragments that, according to the thesis of the paper, emphasize the original scientific achievement of the authors.

Reviewer 5 Report

The paper analyzes the advantages of unmanned electric vehicles with independent drive, braking, and steering. It proposes a control strategy for trajectory tracking that uses fuzzy PI algorithm for longitudinal control and model prediction control theory for lateral control. Fault tolerance is achieved using differential drive and other methods. The proposed strategy improves trajectory tracking accuracy, stability, and safety. In general, this is a good paper dealing with the timely subject.

The following comments and recommendations may be helpful to improve the paper:

Q-1: Begin the introduction by providing some background information on unmanned electric vehicles, their benefits, and challenges associated with their trajectory tracking control. This sets the context for the study and helps readers understand the significance of the proposed trajectory tracking control strategy. Then clearly state the research problem being addressed by the paper, such as the need for effective trajectory tracking control of unmanned electric vehicles. Also, identify the specific research questions being addressed, such as what control algorithm can be used for longitudinal and lateral control, and what methods can be used for fault tolerance.

Q-2: In the introduction, explain how the proposed trajectory tracking control strategy contributes to the field of unmanned electric vehicles, such as by addressing current limitations in trajectory tracking control and providing a more robust and effective solution.

Q-3: It is better to improve the Figure 1, since some words are split and a little messy, and the Figure 4 is not clear. And it is better to enlarge the figures 7, especially the labels.

Q-4: In the fuzzy PI controller design, when you utilize the fuzzy rule in this paper and please try to analyze the fuzzy surface regarding the peak, the valley, and the transition in between. Then why are the four forces of four wheels the same in the Figure 6.

Moderate editing

Reviewer 6 Report

The author proposes an adaptive hierarchical trajectory tracking control method to achieve precise operation of unmanned distributed drive electric vehicles with full-line control. This research also analyzes the vehicle's lateral and longitudinal motion control strategies and the fault-tolerant control strategy of the steering wheel electronic control system. The fault-tolerant control strategy can ensure the safe operation of the vehicle when the steering wheel electronic control system encounters a malfunction. The method is based on model parameter self-adaptive adjustment and hierarchical structure design and has been proven effective through simulation. In addition, the study provides reference materials from relevant literature, which helps readers further understand the related research in this field. The research method is reasonable and has a mathematical model. However, the following points can be improved:

1.  The bibliography should be significantly expanded. The reviewer suggests adding a literature review with the latest references. The latest references are from 2020. It is recommended to add references in MDPI.

2.  The author should emphasize the difference between unmanned full-line control distributed drive electric vehicles and full-line control distributed drive electric vehicles since the control methods, fuzzy PI algorithm, PID controller, and MPC control are common. In other words, the author should emphasize the novelty and contribution of this research.

3.  The author mentioned the fault-tolerant control strategy and illustrated it in Figure 12 on page 16 and the corresponding text description. The strategy ensures the safety and reliability of unmanned full-line control distributed drive electric vehicles by monitoring the driving system, detecting faults in real-time, and addressing them. However, it is not clear under what circumstances real-time faults will occur and the scope of fault-tolerant control.

4.  Figure 13 shows the Fault Detection Strategy Diagram with Chinese characters; please correct them to English.

5. It is not explicitly mentioned whether the proposed adaptive hierarchical trajectory tracking control method can achieve good results in practical applications. Therefore, future research can further explore the feasibility and effectiveness of this method in actual scenarios. Moreover, the paper does not provide specific data and experimental results, so future research can strengthen the experimental part to better verify the effectiveness of the method.

6. It is recommended that the authors' approach increases the limitations of the study in the conclusion.

The English language quality of the paper should be moderately professionally edited in English.

Round 2

Reviewer 4 Report

Since the authors of the manuscript took into account my comments, I propose to accept it for publication.

Reviewer 6 Report

The authors have made corrections to the manuscript based on the reviewers' suggestions. The manuscript has been significantly improved and is now qualified for Applied Sciences.

The English quality is satisfactory, adhering to the standard conventions of academic writing. Only minor enhancements are required to further refine the English quality.